# Widening the Pipeline in Human-Guided Reinforcement Learning with Explanation and Context-Aware Data Augmentation

**Lin Guan**
School of Computing & AI
Arizona State University
Tempe, AZ 85281
lguan9@asu.edu

**Mudit Verma**
School of Computing & AI
Arizona State University
Tempe, AZ 85281
mverma13@asu.edu

**Sihang Guo**
Department of Computer Science
The University of Texas at Austin
Austin, TX 78712
sguo19@utexas.edu

**Ruohan Zhang**
Department of Computer Science
Stanford University
Stanford, CA 94305
zharu@stanford.edu

**Subbarao Kambhampati**
School of Computing & AI
Arizona State University
Tempe, AZ 85281
rao@asu.edu

## Abstract

Human explanation (e.g., in terms of feature importance) has been recently used to extend the communication channel between human and agent in interactive machine learning. Under this setting, human trainers provide not only the ground truth but also some form of explanation. However, this kind of human guidance was only investigated in supervised learning tasks, and it remains unclear how to best incorporate this type of human knowledge into deep reinforcement learning. In this paper, we present the first study of using human visual explanations in human-in-the-loop reinforcement learning (HIRL). We focus on the task of learning from feedback, in which the human trainer not only gives binary evaluative "good" or "bad" feedback for queried state-action pairs, but also provides a visual explanation by annotating relevant features in images. We propose EXPAND (EXPlanation AugmeNted feeDback) to encourage the model to encode task-relevant features through a context-aware data augmentation that only perturbs irrelevant features in human salient information. We choose five tasks, namely Pixel-Taxi and four Atari games, to evaluate the performance and sample efficiency of this approach. We show that our method significantly outperforms methods leveraging human explanation that are adapted from supervised learning, and Human-in-the-loop RL baselines that only utilize evaluative feedback.

## 1 Introduction

Deep reinforcement learning (DRL) algorithms have achieved many successes in solving problems with high-dimensional state and action spaces [27, 3]. However, DRL's performance is limited by its sample (in)efficiency. It is often impractical to collect millions of training samples as required by standard DRL algorithms. One way to tame this problem is to leverage additional human guidance by following the general paradigm of Human-in-the-Loop Reinforcement Learning (HIRL) [51]. It often allows the agent to achieve better performance and higher sample efficiency.

One popular form of human guidance in HIRL is the binary evaluative feedback [15], in which humans provide a "good" or "bad" judgment for a queried state-action pair. This framework allows

35th Conference on Neural Information Processing Systems (NeurIPS 2021).

non-expert humans to provide feedback, but its sample efficiency could be further improved by asking humans to provide stronger guidance. For example, the binary feedback does not tell the agent why it made a mistake. If humans can explain the "why" behind the evaluative feedback, then it is possible to further improve the sample efficiency and performance.

Taking the training of an autonomous driving agent as a motivating example, humans can "explain" the correct action of "apply-brake" by pointing out to the agent that the "STOP" sign is an essential signal. One way to convey this information is through *saliency information*, in which humans highlight the important (salient) regions of the visual environment state. The visual explanation will indicate which visual features matter the most for the decision in the current state. Note that requiring human trainers to provide explanations on their evaluations does not necessarily require any more expertise on their part than is needed for providing only binary feedback. In our driving agent example, the human trainers may not know things like the optimal angle of the steering wheel; however, we can expect them to be able to tell whether an observed action is good or bad, and what visual objects matter for that decision.

In fact, the use of human visual explanation has been investigated by recent works in several supervised learning tasks [39, 35, 34]. They show that the generalization of a convolutional neural network can be improved by forcing the model to output the same saliency map as human - in other words, forces to model to make the right prediction for the right reasons. However, it remains unclear how to effectively incorporate domain knowledge in visual explanation in deep reinforcement learning.

In this work, we present EXPAND, which aims to leverage - EXPlanation AugmeNted feeDback (Fig. 1) to support efficient human guidance for deep reinforcement learning. This raises two immediate challenges (i) how to make it easy for humans to provide the visual explanations and (ii) how to amplify the sparse human feedback. EXPAND employs a novel context-aware data augmentation method, which amplifies the difference between relevant and irrelevant regions

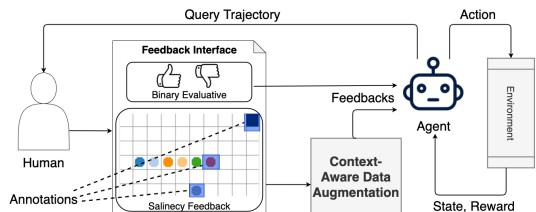

Figure 1: Overview of EXPAND. The agent queries the human with a sampled trajectory for binary evaluative feedback on the action and saliency annotation on the state. The Perturbation Module supplements the saliency explanation by perturbing irrelevant regions. The agent then consumes the integrated feedback and updates its parameters. This loop continues until the agent is trained with feedback queried every $N_f$ episodes. The domain shown for "Visual Explanation" is *Pixel Taxi*.

in human explanation by applying multiple perturbations to irrelevant parts. To reduce the human effort needed to provide visual explanations, EXPAND uses off-the-shelf object detectors and trackers to allow humans to to give explanations in terms of salient objects in the visual state, which are then automatically converted to saliency regions to be used by the RL system. We show that EXPAND agents require fewer interactions with the environment (*environment sample efficiency*) and over 30% fewer human signals (*human feedback sample efficiency*) by leveraging human visual explanation.

We highlight the main contributions of EXPAND below:

- This is the first work that leverages human visual explanation in human-in-the-loop reinforcement learning tasks. We show that EXPAND is the state-of-the-art method to learn from human evaluative feedback in terms of environment sample efficiency and human feedback sample efficiency.

- We benchmark our context-aware data augmentation against standard context-agnostic data augmentation techniques. Our experiment results help shed light on the limitations of existing data augmentations and can benefit future research in accelerating RL with augmented data.

- We show that human visual explanation in a sequential decision-making task can be collected in a low-effort and semi-automated way by using an off-the-shelf object detector and tracker.

We note that EXPAND agents have applicability beyond improving the efficiency of human-in-the-loop RL with sparse environment rewards. In particular, they can be equally useful in accepting human guidance to align the RL system to human preferences. They can be used to incorporate

rewards from human feedback, for example, when environment reward is not defined upfront or to accommodate additional constraints that are not reflected in existing reward signal. EXPAND, thus, can also be useful in aligning objectives of humans in the loop to the agent's [15, 5]. Finally, EXPAND's approach of allowing humans to provide their explanations and guidance in terms of objects of relevance to them, and automatically converting those to saliency regions, is an instance of the general approach for explainable and advisable AI systems that use symbols as a *lingua franca* for communicating with humans, independent of whether they use symbolic reasoning internally [13].

## 2 Related Work

Leveraging human guidance for RL tasks has been extensively studied in the context of imitation learning (learning from demonstration) [37], inverse reinforcement learning [29, 1], reward shaping [28], learning from human preference [7, 22], and learning from saliency information provided by human [49]. Surveys on these topics are provided by [51, 46].

Compared to these approaches, learning from human evaluative feedback has the advantage of placing minimum demand on both the human trainer's expertise and the ability to provide guidance (e.g. the requirements of complex and expensive equipment setup). Representative works include the TAMER framework [15, 45], and the COACH framework [25, 4]. The TAMER+RL framework extends TAMER by learning from both human evaluative feedback and environment reward signal [16, 17]. DQN-TAMER further augments TAMER+RL by utilizing deep neural networks to learn in high dimensional state space [2]. Several approaches have been proposed to increase the information gathered from human feedback, which takes into account the complexities in human feedback-providing behavior. Loftin et al. speed up learning by adapting to different feedback-providing strategies [23, 24]; the Advice framework [11, 6] treats human feedback as direct policy labels and uses a probabilistic model to learn from inconsistent feedback. Although these approaches better utilize human feedback with improved modeling of human behaviors, weak supervision is still a fundamental problem in the evaluative feedback approach.

Human explanatory information has also been explored previously. The main challenge of using human explanation is to translate human high-level linguistic feedback into representations that can be understood by the agents. As an early attempt, Thomaz et al. allow humans to give anticipatory guidance rewards and point out the object tied to the reward [44]. However, they assume the availability of an object-oriented representation of the state. Krening et al., Yeh et al. resort to human explanatory advice in natural language [19, 48]. Still, they assume a recognition model that can understand concepts in human explanation and recognize objects. In this work, we bridge the vocabulary gap between the humans and the agent by taking human visual explanations in the form of saliency maps. Other works use human gaze data as human saliency information, collected with sophisticated eye trackers, to help agents with decision making in an imitation learning setting [50, 52, 14]. They require human data to be collected offline in advance. In contrast, this work is closer to Arakawa et al., Xiao et al., Cui et al., where human trainers are required to be more actively involved during training [2, 47, 8].

Recent works in supervised learning also use human visual explanation to extend the human-machine communication channel [35, 34, 43]. The visual explanation is used as domain knowledge to improve the performance of machine learning models. The way we exploit human explanation via state perturbation can be viewed as a novel way of data augmentation. Data augmentation techniques are widely used in computer vision tasks [21, 40] and have recently been applied to deep RL tasks [18, 20, 33, 26]. The framework *explanatory interactive learning* [43] also leverages human visual explanation with data-augmentation. However, they only focus on classification tasks with low-dimensional representations.

## 3 Problem Setup

We intend to verify the hypothesis that the use of visual explanation and binary evaluative feedback can boost an RL agent's performance. We have an agent $M$ that interacts with an environment $\mathcal{E}$ through a sequence of actions, states, and rewards. Following standard practice in deep RL to make states Markovian, $k$ ($k = 4$) preprocessed consecutive image observations are stacked together to form a state $s_t = [x_{t-(k-1)}, ..., x_{t-1}, x_t]$ [27]. At each time-step $t$, the agent can select an action

from a set of all possible actions $\mathcal{A} = \{1, ..., K\}$ for the state $s_t \in \mathcal{S}$. Then the agent receives a reward $r_t \in \mathcal{R}$ from the environment $\mathcal{E}$. This sequence of interactions ends when the agent achieves its goal or when the time-step limit is reached. This formulation follows the standard Markov decision process framework in RL research [42]. The agent's goal is to learn a policy function $\pi$, a mapping from a state to action, that maximizes the expected return. For a deep Q-Learning agent, the policy $\pi$ is approximated by the Q function $Q(s, a)$, which represents the maximum expected rewards for taking action $a$ at state $s$.

Additionally, we assume a human trainer who provides binary evaluative feedback $\mathcal{H} = (h_1, h_2, ...h_n)$ that conveys their assessment of the queried state-action pairs given the agent trajectory. We define the feedback as $h_t = (x_t^h, b_t^h, x_t, a_t, s_t)$, where $b_t^h \in \{-1, 1\}$ is a binary "bad" or "good" feedback provided for action $a_t$, and $x_t^h = \{Box_1, ..., Box_m\}$ is a saliency map for the image $x_t$ in state $s_t$. $Box_i$ is a tuple $(x, y, w, h)$ for the top left Euclidean coordinates $x$ and $y$, the width $w$, and the height $h$ of the bounding box annotated on the observation image $x_t$.

## 4 Method

EXPAND aims to improve data efficiency and performance with explanation augmented feedback. In the following, we will first describe how EXPAND simultaneously learns from both environment reward and binary evaluative feedback. Then we introduce our novel context-aware data augmentation that utilizes domain knowledge with human explanation. Algorithm 1 in Appendix A presents the train-interaction loop of EXPAND. Within an episode, the agent interacts with the environment and stores its transition experiences. Every few episodes, it collects human feedback queried on a trajectory sampled from the most recent policy. All the human feedback is stored in a feedback buffer similar to the replay buffer in off-policy DRL [27]. The weights of RL agent are updated twice, first using sampled environment data as in usual RL update, and then using human feedback data, in a single training step. In our experiment, both EXPAND and the baselines follow the same train-interaction loop.

### 4.1 Learning from Environment Reward and Binary Feedback

The underlying RL algorithm of EXPAND can be any off-policy Q-learning-based algorithm. In this work, we use Deep Q-Networks [27] combined with multi-step returns, reward clipping, soft-target network updates and prioritized experience replay [38]. We refer to this RL approach as Efficient DQN.

To learn from binary evaluative feedback, we propose a new method that doesn't require additional parameters to explicitly approximate human feedback. We use the advantage value to formulate the feedback loss function, called *advantage loss*. Advantage value is the difference between the Q-value of the action upon which the feedback was given and the Q-value of the current optimal action calculated by the neural network. Given the agent's current policy $\pi$, state $s$, and action $a$ for which the feedback is given, the advantage value is defined as:

$$A^\pi(s, a) = Q^\pi(s, a) - V^\pi(s) = Q^\pi(s, a) - Q^\pi(s, \pi(s)) \tag{1}$$

Hence, the advantage value quantifies the possible (dis)advantage the agent would have if some other action were chosen instead of the current-best. It can be viewed as the agent's judgment on the optimality of an action. Positive feedback means the human trainer expects the advantage value of the annotated action to be zero. Therefore, we define a loss function, i.e., the advantage loss, which forces the network to have the same judgment on the optimality of action as the human trainer. Intuitively, we penalize the policy-approximator when a marked "good" action is not chosen as the best action, or when a marked "bad" action is chosen as the best action.

For a feedback $h = (x^h, b^h, x, a, s)$, when the label is "good", i.e., $b^h = 1$, we expect the network to output a target value $\hat{A}(s, a) = 0$, so the loss can be defined as $|\hat{A}(s, a) - A^\pi(s, a)| = Q^\pi(s, \pi(s)) - Q^\pi(s, a)$. When the label is "bad", i.e., $b^h = -1$, we expect the network to output an advantage value $A^\pi(s, a) < 0$. Since here we do not have a specific target value for $A^\pi(s, a)$, we resort to the idea of large margin classification loss [31], which forces $Q^\pi(s, a)$ to be at least a margin $l_m$ lower than the Q-value of the second best action, i.e., $\max_{a' \neq a} Q^\pi(s, a')$. One advantage of this interpretation of human feedback is that it directly shapes the Q-values with the feedback information and does not require additional parameters to model human feedback. This kind of large margin loss has been shown to be effective in practice by previous works like DQfD [12].

Formally, for human feedback $h = (x^h, b^h, x, a, s)$ and the corresponding advantage value $A_{s,a} = A^\pi(s,a)$, the advantage loss is:

$$L_A(s,a,h) = L_A^{Good}(s,a,h) + L_A^{Bad}(s,a,h) \tag{2}$$

where

$$L_A^{Good}(s,a,h;b^h=1) = \begin{cases} 0 & ; A_{s,a} = 0 \\ Q^\pi(s,\pi(s)) - Q^\pi(s,a) & ; \text{otherwise} \end{cases} \tag{3}$$

and,

$$L_A^{Bad}(s,a,h;b^h=-1) = \begin{cases} 0 & ; A_{s,a} < 0 \\ Q^\pi(s,a) - (\max_{a' \neq a} Q^\pi(s,a') - l_m) & ; A_{s,a} = 0 \end{cases} \tag{4}$$

Note that the COACH framework [25] also interprets human feedback as the advantage function. In COACH's formulation, the advantage value is exactly the advantage term in the policy gradient equation–human trainers are supposed to provide positive/negative policy-dependent feedback only when the agent performs an action better/worse than that in current policy. Thus, such a formula is restricted to on-policy policy-gradient methods. In contrast, the advantage function in EXPAND aims to capture the relative utility of all the actions. Here human feedback is direct policy advice as in the Advice framework [11], indicating whether an action is preferable regardless of the agent's current policy. This property makes the advantage loss in EXPAND a better fit for off-policy value-based methods.

## 4.2 Leveraging Human Visual Explanation

Human visual explanation informs the agent about which parts of the state matter for making the right decision. These "parts" of the state could be specific regions or objects. In EXPAND, each saliency map consists of a set of bounding boxes over the images, marking a region's importance in making the decision. The intuition is that the agent's internal representation must correctly capture the relevant features before it can make an optimal decision. Based on this, we propose a novel context-aware data augmentation technique, which applies multiple perturbations to irrelevant visual regions in the image and forces the model to be invariant under these transformations. The key idea here is that, the manipulations to irrelevant regions should not affect the agent's policy.

To get more insights on how our data augmentation method benefits policy learning, we follow the causal-graph interpretation of representation learning with data augmentation visualized in Fig. 2a [26]. In this model, each image observation $X$ can be expressed by *content* variables and *style* variables. Content variable $C$ contains all the necessary information to make the correct prediction, while style variable $S$ contain all other information that doesn't influence current downstream task. That said, only content is causally related to current downstream target of interest $Y$, and content $C$ is independent of style $S$. The goal of representation learning is to accurately approximate the invariant part (content) and ignore the varying parts (style). Based on this formulation, data augmentation is essentially a way to emulate style variability by performing interventions on the style variables $S$. However, since the choice of data augmenta-

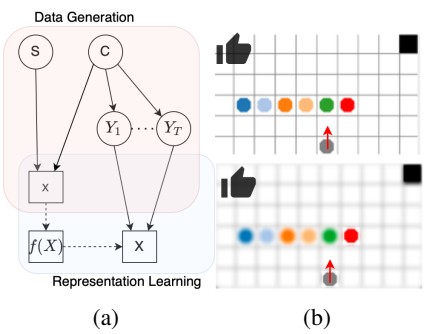

Figure 2: (a) Modeling representation learning problem with data augmentation using a causal graph. Figure from [26]. (b) Example state of Pixel-Taxi, the top image represents the original state and the bottom represents a sample augmented state.

tions implicitly defines which aspects of the data are designated as style and which are content, to make sure the context information $C$ is preserved, existing data augmentation methods in RL are limited to conservative transformations such as random translation or random cropping. Previous works show that transformations altering content $C$ can even be detrimental, which is referred to as aggressive augmentations in [32].

Different from standard data augmentations in RL, prior domain knowledge about state context is provided by human explanation in EXPAND. Hence, we can resort to a wider range of transformations, and thereby more informatively highlight the content variable $C$ by applying various transformations only to regions marked as irrelevant (style variable $S$) while keeping the relevant parts (content variable $C$) unchanged. Figure 2b shows an example state from Pixel-Taxi domain where the top image is the original state having gray cell as the taxi, red as the passenger, and black as the destination. The bottom image shows an example image used by EXPAND which contrastively highlights the taxi and passenger by performing a Gaussian blur over the remainder of the image observation. This illustrates why the proposed context-aware augmentation can be more informative than standard data augmentations.

We choose to use Gaussian blurring to perturb the irrelevant regions, which is also used in a previous Explainable RL work [10]. The reason we use Gaussian blurring is that, it effectively perturbs objects in image while does not introduce new information. One counterexample here can be, transformations like *cutout* might significantly change state content in environments where small black blocks have particular semantic meaning.

Formally, consider a feedback $h = (x^h, b^h, x, a, s)$, we need to convert state $s$ into augmented states $f(s)$ with perturbations on irrelevant regions. Let $\mathbb{M}(x, i, j)$ denote a mask over relevant regions, where $x(i, j)$ denotes the pixel at index $(i, j)$ for image $x$. We then have:

$$\mathbb{M}(x, i, j) = \begin{cases} 1 & \text{if (i, j) lies in Box, } \exists \text{ Box} \in b^h \\ 0 & \text{otherwise} \end{cases} \tag{5}$$

$$\phi(x, M, i, j) = x \odot (1 - \mathbb{M}(x, i, j)) + G(x, \sigma_G) \odot \mathbb{M}(x, i, j) \tag{6}$$

Then we can perturb pixel $(i, j)$ in image observation $x$ according to mask $\mathbb{M}$ using a function $\phi$ defined as above, where $\odot$ is the Hadamard Product and function $G(x, \sigma_G)$ is the Gaussian blur of the observation $x$. Hence, we can get an augmented state $f(s)$ with perturbed irrelevant regions by applying $\phi(x, M)$ to each stacked frame $x$ with the corresponding mask $\mathbb{M}$.

We use the context-aware data augmentation in the following two ways:

1. For any human feedback $h = (x^h, b^h, x, a, s)$, we train the model by calculating the advantage loss with the original data $h$ as well as the augmented feedback $h' = (x^h, b^h, x, a, f(s))$. Note that one human feedback can be augmented to multiple feedbacks by varying the parameters of $f(s)$. In EXPAND, we use Gaussian perturbations of various filter sizes and variances (see Appendix B for detailed settings).

2. To encourage the RL model's internal representation to accurately capture relevant visual features and ignore other irrelevant parts, we enforce an explicit *invariance constraints* via an auxiliary regularization loss:

$$L_I = \frac{1}{g} \sum_{i=1}^{g} \frac{1}{|\mathcal{A}|} \sum_{a \in \mathcal{A}} \|Q(s, a) - Q(f_i(s), a)\|_2 \tag{7}$$

where $\mathcal{A}$ is the action set and $g$ is the number of perturbations.

### 4.3 Combining Feedback and Explanation Losses

We linearly combine all the losses to obtain the overall feedback loss:

$$L_F = \lambda_A L_A + \lambda_I L_I \tag{8}$$

where $\lambda_A$ and $\lambda_I$ are the weights of advantage loss and invariant loss respectively. In all experiments, we set $\lambda_A$ to 1.0 without doing hyper-parameter tuning. For $\lambda_I$, we set it to 0.1 as suggested in [33], which also applies an invariant constraint on the RL model but with standard data augmentations. The agent is trained with usual DQN loss as well as $L_F$. Note that in EXPAND, the advantage loss is computed with both original human feedback and augmented human feedback. For a baseline agent that doesn't utilize human visual explanation (only trained with usual DQN loss and advantage loss with original human feedback), we refer to it as DQN-Feedback.

### 4.4 Collecting Human Visual Explanation

Though human visual explanation offers a strong way to make human-agent interaction more natural and effective, it might impose an excessive amount of workload if we require the trainer to annotate the image for every single query. To account for possible overheads and reduce human effort, we design an object-oriented interface that enables the human to provide guidance by effortlessly pointing out the labels of salient objects. Note that this design is in line with the idea of using a "symbolic" interface in human advisable AI, which argues that humans are more comfortable with symbol-level communication, and the agents are free to learn their own internal representation as long as the user interface can ground the human advice (e.g. to pixel space) [13]. Technically, our object-oriented interface contains a tracking and detection module, with which the human trainers only need to annotate at the beginning or when the tracker/detector works imperfectly. For example, in the car driving game Enduro (Fig. 3), all the lanes and cars are automatically highlighted and tracked, so the human trainers only need to deselect irrelevant objects in the image.

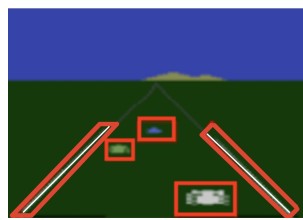

Figure 3: An example of object detection in car driving game Enduro. Users were allowed to remove or add other bounding regions at will.

A screenshot of our video-player like interface can be found in Appendix F. Trainers can start/pause the replay of the queried trajectory and provide explanations by directly drawing bounding boxes on the screen. The trainers can also adjust the "video" frame rate based on their needs. Similar to Deep-TAMER [45], we also applied each received feedback and explanation to frames that are displayed between 2 and 0.2 seconds before the feedback occurred – we assume that within this 1.8-second window, the salient regions/objects should be the same.

## 5  Experimental Evaluation

The experimental evaluation of the work would try to answer the following questions.

1. Whether the use of human explanation improves the environment and feedback sample efficiency?
2. Does EXPAND utilize the human explanation better than other baselines?
3. Is context-aware data augmentation more informative than standard data augmentation?

To answer the first question, we compare EXPAND to DQN-Feedback and an HIRL algorithm DQN-TAMER which combines TAMER with deep RL [2].

For the second question, we compare EXPAND with two other explanatory interactive learning methods that are adapted from supervised learning, namely Ex-AGIL and Attention-Align. Ex-AGIL is adapted from AGIL [50], which trains a separate attention prediction network to generate visual explanation for unseen states. The predicted attention is used as a mask to filter out irrelevant information in the image observation. Then the masked state is fed as input to the policy network. Attention-Align uses an auxiliary *attention alignment loss* that is similar to the loss functions for supervised learning in [36, 39, 34, 35]. It penalizes if the agent's attention heatmap does not align with human visual explanation. To efficiently obtain the differentiable attention heatmap of the model, we use an Explainable RL algorithm FLS [30]. The saliency map produced by FLS-DQN is essentially the model's prediction on whether a pixel should be included in a bounding box. Hence the attention alignment loss here is defined to be the mean square error between agent's prediction and human visual explanation. The implementation details of the baselines can be found in Appendix C.

Finally, to answer the third question, we replace the context-aware data augmentation in EXPAND with standard augmentations that do not use human saliency information. The context-agnostic augmentations we compare to include Gaussian blurring and random cropping, which result in state-of-the-art sample efficiency in recent works [20, 18]. Note that EXPAND only augments states which were queried to the human trainer, hence for this comparison, the context-agnostic methods only augment the states for which the system received a human feedback.

We conducted experiments on three tasks: Pixel-Taxi (Fig. 2b) and four Atari games. The Pixel-Taxi domain is similar to the Taxi domain which is widely used in RL research [9]. It is a grid-world

setup in which the taxi agent, occupies one grid cell at a time, and passengers (denoted by different colored dots) occupy some other cells. The taxi agent's goal is to pick up and transport the correct passenger to the destination. To force the agent to learn passenger identities instead of memorizing their "locations", we randomize the passengers' positions at the beginning of each episode. A reward is given only when the taxi drops off the correct passenger at the destination cell. In addition, we choose four Atari games with default settings: Pong, Asterix, Montezuma's Revenge, and Enduro. Original Enduro can be infinitely long and can make human training impractical. Therefore, in Enduro, our goal is to teach the agent to overtake as many cars as possible within 1000 environment steps (an episode); hence we denote this task as Enduro-1000. In Montezuma's Revenge, we train the agent to solve the first room within 1000 environment steps per episode; hence we denote this task as MR Level 1 or simply MR.

In all experiments, both EXPAND and the baselines use Efficient DQN as the underlying RL approach. Efficient DQN uses the same DQN network architecture designed by Mnih et al. [2015]. Details on the architecture and hyperparameters can be found in Appendix E. Following the standard pre-processing [27], each frame is converted from RGB format to grayscale and is resized to $84 \times 84$. The input pixel values are normalized to be in the range of $[0, 1]$. During training, we start with an $\epsilon$-greedy policy ($\epsilon = 1.0$) and reduce $\epsilon$ by a factor of $\lambda_\epsilon$ at the end of each episode until it reaches 0.01. All the reported results are averaged over 5 random seeds.

Algorithm 1 describes the steps for obtaining human feedback: for every $N_f$ ($N_f = 4$) episode, we sample one trajectory and query the user for binary evaluative feedback as well as a visual explanation. Active querying, although preferable, is left for future experiments since the goal of this work is to demonstrate whether human explanations can effectively augment binary feedback. When collecting feedback, we allow humans to watch the queried trajectories and provide feedback at will (the human can choose not to provide feedback for some queried states).

## 5.1 Evaluation using Synthetic Feedback and Explanation from Oracle

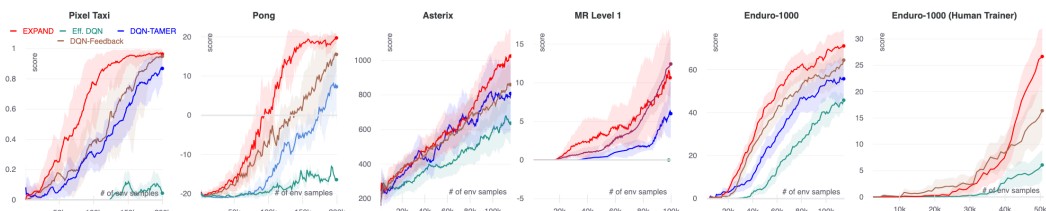

Figure 4: The smoothed learning curves of EXPAND and the baselines. The solid lines show the mean score over 5 random seeds. The shaded regions represent the standard error of the mean. In Pixel-Taxi, the score is a running average over the last 20 rollouts. Our method (EXPAND, in red) outperforms the baselines in all the tasks.

To perform a systematic analysis, we first conducted experiments that include 5 runs of all algorithms with a synthetic oracle. A synthetic oracle allows us to run a large number of experiments and give consistent feedback across different runs, providing fair and systematic comparisons between different approaches [11, 2]. The oracle uses trained models from *Atari Zoo* [41] for the Atari games, and a trained DQN model for Pixel-Taxi. To annotate the "relevant" regions on the image observation, we use hand-coded models to highlight the taxi-cell, the destination-cell, and the target-passenger in Pixel-Taxi; the two paddles and the ball in Atari-Pong; the player vehicle, lanes, and other vehicles in front of the agent in Enduro-1000. In both MR and Asterix the oracle highlighted the agent and other regions like, monster, ladder and key in MR, and enemies and target in Asterix, depending upon whether they are spatially close to the agent. (See Fig. 3 and Appendix D for other examples).

**Improvements on Environment and Feedback Sample Efficiency :** Fig. 4 compares the environment sample efficiency as well as performance between EXPAND (in red) and the baselines. To answer the first question posed in Section 5, the plot clearly shows that EXPAND outperforms the HIRL baselines (DQN-Feedback and DQN-TAMER) by a large margin across all the tasks except MR, where EXPAND was consistently at par with DQN-Feedback. In Pixel-Taxi and Pong, EXPAND is able to learn a near optimal policy with 35% less environment samples/ feedback samples. In Enduro-1000, EXPAND manages to consistently obtain a score over 60 by using 80k samples compared to 120k samples used by DQN-TAMER, an over 30% improvement. In Asterix and MR,

EXPAND achieved a considerably higher score than DQN-TAMER. Finally, since human feedback is obtained at fixed intervals, an improvement in environment sample efficiency (x axis in the figure 4) would imply subsequent improvement in feedback sample efficiency.

**EXPAND versus other Explanatory Interactive Learning methods :**

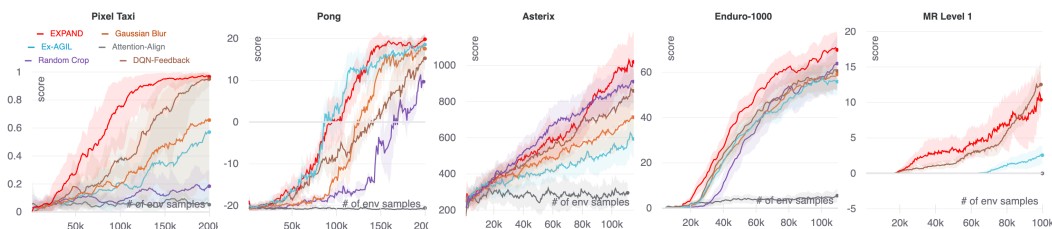

Figure 5: Comparison of EXPAND with other perturbation techniques and attention-based Interactive Learning methods.

As mentioned earlier, to answer the second question posed in Section 5, we compare EXPAND with Ex-AGIL and Attention-Align. From Fig. 5, we can observe that Ex-AGIL only improves the baseline DQN-Feedback in Pong, while in other more visually-complex environments, the auxiliary attention prediction harms the performance. This highlights two issues with approximating human attention with an additional model, first, the approximation may fail for unseen states and second, the visual explanation is not used as a strong signal to directly regularize the policy network. On the other hand, the second baseline Attention-Align fails to learn a usable policy in all five tasks. A potential reason for its failure is a misalignment between the attention prediction objective and the reward-seeking objective. This indicates that this type of attention alignment loss might not be suitable for a less stable learning system like deep RL. In contrast, data-augmentation methods for RL have been empirically examined over the years, hinting that EXPAND's methodology is more stable.

**EXPAND versus Context-Agnostic Data Augmentations :** Additional comparisons between EXPAND and standard data augmentations were made to verify our hypothesis that context-aware data augmentation can be more informative than context-agnostic augmentations. As expected, Gaussian blurring and random cropping fail to help the agent (Fig. 5). Interestingly, the two context-agnostic data augmentations even degrade the performance in some tasks since they add unnecessary complexities when the agent tries to infer human trainer's intent behind the binary evaluative feedback. This contradicts EXPAND's methodology, which contrastively highlights relevant regions. This result suggests that standard data augmentation helps RL by obtaining more data to prevent overfitting [18], but its informativeness can be further improved by incorporating domain knowledge as in EXPAND.

## 5.2 Evaluation with Human in the Loop

So far, we have presented our results using synthetic oracles. In this section, we will present results with a human trainer. We run this experiment on Enduro-1000. The objective of this user study is to address the following problems:

1. Can EXPAND perform well with feedback and explanation from human trainer, considering human feedback can be sparser and noisier than synthetic feedback?

2. Is there a low-cost way to collect human visual explanation in sequential decision tasks?

In the experiment, the agent queried the human at an interval of every 10 episodes. We limit the annotation time for each query to be around 5 minutes. Hence the total interaction time is at most 30 minutes. Each algorithm was run 3 times with different random seeds.

Within the 30-minute interaction time, in the baseline DQN-Feedback, human trainers only need to give binary evaluative feedback, and the trainers provided 2405 binary feedbacks on average. In EXPAND, the human trainers provided 2026 feedback-explanation pairs on average within the same time limit. The difference in the number of feedbacks is not large so it suggests that the cost of providing visual explanation is low. From Fig. 4, we can observe that EXPAND significantly outperforms DQN-Feedback in the user study, hinting that the use of human explanation can also improve sample complexity given the same wall-clock interaction time.

## 5.3 Ablation Study

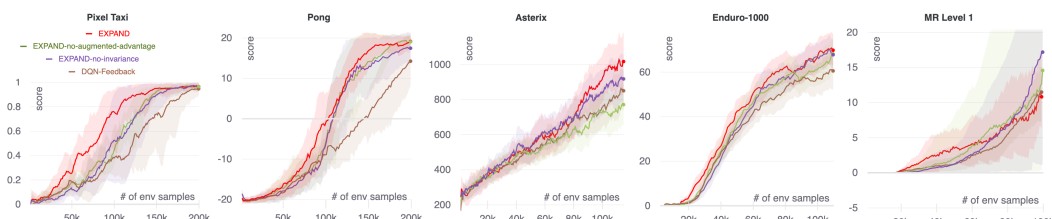

Figure 6: Ablation experiments analyzing the effects of each individual loss terms on the performance. The results verify that the combination of both loss terms leads to the best performance.

To get more insights into the losses in EXPAND, we conducted an ablation study on two variants of EXPAND, which either only use the augmented data to compute the invariance loss (EXPAND-no-augmented-advantage) or only use the augmented data in the advantage loss (EXPAND-no-invariance). Fig. 6 shows that both variants of EXPAND eventually converged to near-optimal policies. We note that EXPAND-no-invariance (in purple) and EXPAND-no-augmented-advantage (in light green) perform significantly better than DQN-Feedback, highlighting that each saliency loss alone provides significant improvements over the baseline. Finally, combining these losses (EXPAND, in red) boosts this performance even further, indicating that combining invariance constraint with advantage loss with augmented data is a better approach.

## 6   Conclusion & Future Work

In this work, we presented a novel method to integrate human visual explanation with their binary evaluations of agents' actions in a Human-in-the-Loop RL paradigm. We show that our proposed method, EXPAND, outperforms previous methods in terms of environment sample efficiency, and shows promising results for human feedback efficiency. Future work can experiment with different types of perturbations beyond Gaussian blurs, including domain-dependent perturbations that involve object manipulation. Finally, as we mentioned at the end of introduction, EXPAND agents are an example of AI agents that can take human advice in terms of a *symbolic lingua franca* [13], and are a promising approach for the more general problem of human-advisable RL systems that can be aligned to human preferences through symbolic advice.

## Acknowledgement

This research is supported in part by ONR grants N00014- 16-1-2892, N00014-18-1- 2442, N00014-18-1-2840, N00014-9- 1-2119, AFOSR grant FA9550-18-1-0067, DARPA SAIL-ON grant W911NF-19- 2-0006 and a JP Morgan AI Faculty Research grant. Ruohan Zhang's work on this paper was done when he was a PhD student at The University of Texas at Austin.

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
