# A  The Overall Workflow of EXPAND

---
**Algorithm 1:** Train - Interaction Loop
---

**Result:** Trained Eff. DQN agent M
**Input:** Eff. DQN agent M with randomly-initialized weights $\theta$, replay buffer $\mathcal{D}$, human feedback buffer $\mathcal{D}_h$, feedback frequency $N_f$, total episodes $N_e$, maximum number of environment steps per episode $T$, update interval b
**Begin**
**for** $i = 1$ **to** $N_e$ **do**
  **for** $t = 1$ **to** $T$ **do**
    Observe state s;
    Sample action from current policy $\pi$ with $\epsilon$-greedy, observe reward r and next state s$'$ and store (s, a, r, s$'$) in $\mathcal{D}$;
    **if** $t \bmod b == 0$ **then**
      Sample a mini-batch of transitions from $\mathcal{D}$ with prioritization;
      Perform standard DQN update with sampled environment data;
      Sample a mini-batch of human feedback from $\mathcal{D}_h$;
      Update Eff. DQN with human feedback data;
    **end if**
  **end for**
  **if** $i \bmod N_f == 0$ **then**
    Obtain the last trajectory $\tau$ from $\mathcal{D}$;
    Query $\tau$ to obtain feedback $\mathcal{H}_i$;
    Append $\mathcal{H}_i$ to buffer $\mathcal{D}_h$;
  **end if**
**end for**
**End**

---

# B  Settings of Gaussian Blurring in EXPAND

In EXPAND, we augment each human evaluated state to 5 states. To verify 5 is sufficient, we also experimented with the numbers of augmentations required in each state to get the best performance. Figure 7 shows a comparison when the number of augmentations is varied among $\{1, 5, 12\}$ for Pixel Taxi and Pong using a synthetic oracle. The plots suggest that increasing augmentations only evoke slight performance gains, and therefore setting the number of perturbations to 5 for EXPAND is apt. The settings of the Gaussian blurring filters are listed below:

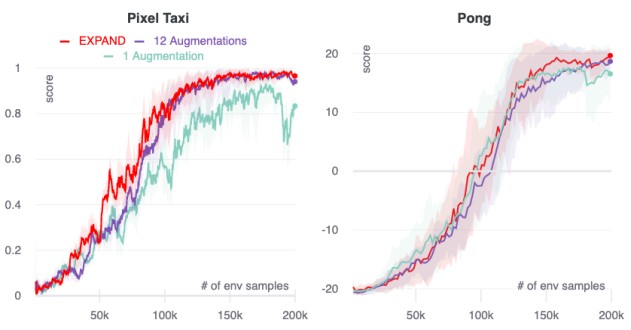

Figure 7: Learning curves of the variants of EXPAND with different number of augmentations.

- 1 Augmentation:
  - filter size: 5, $\sigma$: 5
- 5 Augmentations:
  - filter size: 5, $\sigma$: 2
  - filter size: 5, $\sigma$: 5
  - filter size: 5, $\sigma$: 10
  - filter size: 11, $\sigma$: 5
  - filter size: 11, $\sigma$: 10

- 12 Augmentations:
  - filter size: 5, $\sigma$: 2
  - filter size: 5, $\sigma$: 5
  - filter size: 5, $\sigma$: 10
  - filter size: 7, $\sigma$: 3
  - filter size: 7, $\sigma$: 5
  - filter size: 7, $\sigma$: 10
  - filter size: 9, $\sigma$: 3
  - filter size: 9, $\sigma$: 5
  - filter size: 9, $\sigma$: 10
  - filter size: 11, $\sigma$: 3
  - filter size: 11, $\sigma$: 5
  - filter size: 11, $\sigma$: 10

# C  Implementation Details

## C.1  Ex-AGIL

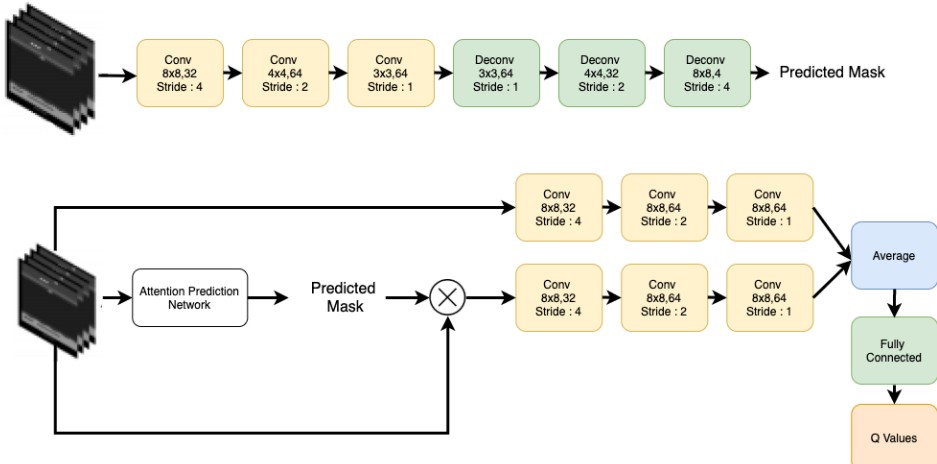

Figure 8: AGIL network architectures. The upper one is the attention prediction network, and the bottom one is the policy network.

AGIL [50] was designed to utilize saliency map collected via human gaze. In this work, human salient information is provided by annotated visual explanation. Though the source of explanatory information is slightly different, we can expect that AGIL with some minor changes (called Ex-AGIL) should also work well when the gaze input is replaced with visual explanation. Note that we don't aim to compare the two different ways to gather human salient information, but rather focus on which method better utilizes human visual explanation. The network architectures are shown in Fig. 8. Two modifications were made to AGIL: firstly, rather than training the attention network and policy network sequentially as in supervised behavioral cloning, the attention network and policy network are both updated on every DQN update step. Secondly, unlike human gaze, human annotated saliency map (bounding boxes) doesn't induce any probability distribution. Hence, we view the output of attention network as the prediction of whether a pixel should be included in a human annotated bounding box. Accordingly, to train the network, we applied a simple mean-squared error between the predicted saliency maps and human visual explanations, similar to the explanation loss terms in [34, 39]:

$$L_{explanation} = \frac{1}{N} \sum_{i=1}^{N} (A_i - e_i)^2 \tag{9}$$

where $e_i$ is the predicted saliency mask for state $s_i$ and $A_i$ is the preprocessed (resized and stacked) human explanation binary mask, in which 1 means the pixel is highlighted in human explanation. The policy network is trained with the same losses as DQN-Feedback (standard DQN loss + advantage

feedback loss), except that the states are masked with the predicted saliency map in the way depicted in Fig. 8.

## C.2 Attention-Align

Different from Ex-AGIL, Attention-Align aims to leverage human visual explanation without training a separate saliency prediction network. It uses the same mean-squared error loss in Ex-AGIL but the $e_i$ is obtained with some interpretable reinforcement learning method. Here, we choose to use the method FLS (Free Lunch Saliency) [30], which produces built-in saliency maps by adding a self-attention module that only allows selective features to pass through. More specifically, to get the agent explanation $e_i$, FLS applies transposed convolution to the neuron activations of the attention layer. This allows us to more efficiently compute the explanation loss compared to other methods such as computing Jacobian of the input image or bilinear upscaling of the attention activations.

The Eff. DQN is then jointly trained with standard DQN loss, feedback loss (advantage loss), and the explanation loss. Note that the weight of explanation loss is set to 0.1 as suggested in previous works [34, 39].

## C.3 Context-Agnostic Data Augmentation

We compared EXPAND to two context-agnostic data augmentations, namely random cropping and Gaussian blurring. Random cropping pads the four sides of each $84 \times 84$ frame by 4 pixels and then crop back to the original $84 \times 84$ size. Gaussian blurring applies a $23 \times 23$ square Gaussian kernel with standard deviation sampled uniformly in $(2, 10)$. Note that in EXPAND we use a fixed set of Gaussian filters, which can be more efficient with respect to wall-clock time. We also experimented with context-agnostic Gaussian blurring with fixed filters. Our result shows there is no significant performance difference between fixed filters and randomly sampled filters.

# D  Synthetic Feedback and Explanation

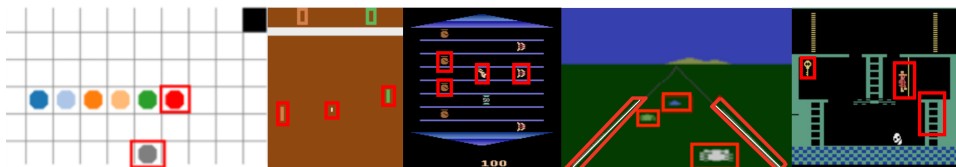

Figure 9: Examples of visual explanations for (left to right) Pixel-Taxi, Pong, Asterix, Enduro-1000, and MR Level 1.

We evaluated EXPAND against the baselines using an oracle. To get the synthetic visual explanation, we use hard-coded program to highlight the relevant regions. In Pixel Taxi, the relevant objects are the agent (gray block), passenger (red block) and the destination (black block). In Pong, we highlight the ball and the two paddles. In Asterix, the closest target(s) and the enemies in nearby lane(s) are highlighted. In Enduro-1000, we highlight the lanes, the player car, and cars in front of the player. In MR, candidate regions/objects include the agent, monster, key, platforms, and ladders. Certain regions will be highlighted depending on the agent's location. Examples of synthetic visual explanations can be found in the Fig. 9.

# E  Hyper-parameters

- Convolutional channels per layer: [32, 64, 64]
- Convolutional kernel sizes per layer: [8, 4, 3]
- Convolutional strides per layer: [4, 2, 1]
- Convolutional padding per layer: [0, 0, 0]
- Fully connected layer hidden units: [512, number of actions]
- Update interval: 4

- Discount factor: 0.99
- Replay buffer size: 50,000
- Batch size: 64
- Feedback buffer size: 50,000
- Feedback batch size: 64
- Learning Rate: 0.0001
- Optimizer: Adam
- Prioritized replay exponent $\alpha = 0.6$
- Prioritized replay importance sampling exponent $\beta = 0.4$
- Advantage loss margin $l_m = 0.05$
- Rewards: clip to [-1, 1]
- Multi-step returns: $n = 5$
- $\epsilon$ episodic decay factor $\lambda_\epsilon$: 0.99 in Pixel Taxi, 0.9 in Atari games

## F   User Study

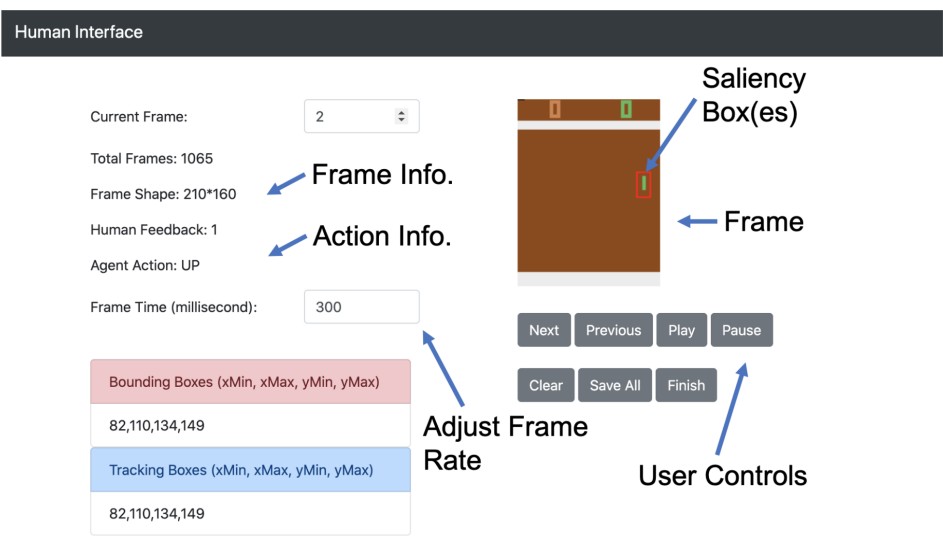

Figure 10: Web Interface to collect saliency and binary feedback from human trainers. This example uses a frame from Pong.

Fig 10 shows the web-interface used to conduct the user experiment with three graduate students as the human subjects. Before the experiment began, the trainers were briefed about the usage of this interface. The trainers were asked to provide their consent to use any information they provide for any analysis and experiments.

The feedback interface, Fig 10, is divided into two columns, the information pane (left) and the control pane. (right). On the information pane, users were able to adjust the frame rate at which they would want to view the "video" made using the state transitions. Users could also view bounding-box related information, such as which ones were the drawn boxes and the boxes suggested by the implemented object tracker. On the right pane, users could view the frame upon which they would give their saliency feedback. Users could see the agent's current action in the information pane. For their ease, we provided controls like *Pause*, *Play*, skip to *Next* Frame, go to *Previous* Frame, *Clear* all bounding boxes on the frame, *Save* feedback and finally *Finish* the current feedback session. To provide a binary feedback on agents' actions, users were required to press keyboard keys "A" for a good-feedback, "S" for a bad-feedback, and "D" for no feedback. They could provide saliency explanations via clicking at the required position on the image frame and dragging to create a rectangular selection.