# OpenReview forum: "Widening the Pipeline in Human-Guided Reinforcement Learning with Explanation and Context-Aware Data Augmentation"
_NeurIPS.cc/2021/Conference — NeurIPS 2021 Spotlight_

### Official Review · Reviewer_1gtV · 2021-07-13

**Rating:** 7
**Confidence:** 3

**Summary:**

They propose methods to better learn (off-policy) Q functions using binary human feedback, and also using visual explanations from humans.

**Limitations And Societal Impact:**

not much here

**Main Review:**

(disclosure:  I had briefly seen a previous version of this as a poster under a different name.  I do not know the authors and don't think it affects my review)

I like the direction of this work.  It seems good generally for humans to try to indicate subsets of observations that most affect actions as a form of feedback.  Overall, I am somewhat convinced it is a promising method, although I think the presentation of the results could be a lot better.  I would've liked to see experiments in a harder environment where I think the gap between EXPAND and DQN-feedback could've been even larger.  Also would've been nice to see at least a picture of each environment and the oracle behavior, given the nature of the work.

Figure 2: I kinda feel looking at this picture that cutout would be more natural.  would that have been worth trying?

Baselines:
- DQN-TAMER is never really explained.  It would be nice to explain the difference between it and DQN-feedback, so the reader doesn't need to look at other papers and think about it
- Not super important, but could context-aware data augmentation be applied to DQN-tamer as another baseline?
- another baseline to get an understanding of how much augmentation helps, is to run DQN-feedback, but training on an extra frames for each would've-been-augmented frame.  presumably this is a strong baseline, and it would be more impressive (but plausible) to beat it.  if I understand correctly, this would also control better for amount of compute used

Improvements on Environment and Feedback Sample Efficiency
- it would be good to know what percentage of frames get augmented.

Comparison to explanatory interactive learning methods:  this section overall felt unconvincing to me
- Attention-Align seems a little complex and the results seem very poor, making me wonder if there were implementation issues or if it was otherwise a poor choice of baseline for some unknown reason.  I'm guessing there's a hyperparameter to tune to balance the attention prediction objective, for example?
- The comparison of EXPAND with Ex-AGIL seems like it could easily come down to number of oracle/human samples + efficiency of learning the mask, since in one case we have an additional learned model.  If the model is good, then it seems like you get to do a lot more augmentation, right?
- why were only two environments used in figure 5?  also line 374 says "all three tasks" when there are just two
- might be nice for Figure 5 to have a no-aug baseline

EXPAND versus Context-Agnostic Data Augmentations
- why not use the augmentations on all states, if there is an oracle?
- it would be good to know what percentage of frames get augmented.  it would be interesting to see, e.g. 100% gaussian blur vs. gaussian blur except when there's human data
- overall comparisons here look solid, although again I wonder why there are only 2 environments

5.2
- the human trainer seems to do worse than the synthetic oracle.  is this just because the oracle is quite good?  it might be interesting to find a harder task where it's not as easy to write a synthetic oracle - I think results would be more compelling in such a setting.
- why was DQN-tamer not included in this?

5.3
- i assume these were done with the oracle again.  it would be good to explicitly state this
- again, I wonder why the other environments weren't included here

**Time Spent Reviewing:**

2

---

> ### Author Response · Authors · 2021-08-10
> **Response to Review**
>
> We thank the reviewer for their comments and suggestions, here are some clarifications on some of the questions raised:
>
>
> ### 1. Is EXPAND sufficiently evaluated? More experiments in harder environments where the gap between EXPAND and DQN-feedback is even larger?
>
> While any system can benefit from further experiments, we do believe that our evaluation of EXPAND is comprehensively covering the largest set of environments with various types of games and different levels of task complexity. For the reviewers’ reference, here we list the environments used in some previous learning-from-human-feedback works: Deep TAMER [1] uses only one Atari environment Bowling; DQN-TAMER [2] was evaluated on two gridworld-like environments (like our Taxi environment), and recent work FRESH [3] was only tested in Bowling and Skiing. Hence, the results from our existing environments should be sufficient to show EXPAND is a promising approach.
>
> We agree that it can be interesting to find what are the most complex tasks that EXPAND can scale to. But we would argue that, as an initial attempt to incorporate human attention prior into a deep RL model, the over 30% reduction in environment samples & feedback samples is already a notably large “performance gap”, which suggests adding human multi-modal feedback is the right direction to improve the data efficiency of interactive reinforcement learning, and thereby scaling it to more complex tasks. We believe the value of this paper lies more in showing the effectiveness of utilizing richer human feedback and encouraging more efforts to be made in this direction.
>
> [1] Warnell, Garrett, et al. "Deep TAMER: Interactive agent shaping in high-dimensional state spaces.”
>
> [2] Arakawa, Riku, et al. "Dqn-tamer: Human-in-the-loop reinforcement learning with intractable feedback."
>
> [3] Xiao, Baicen, et al. "Fresh: Interactive reward shaping in high-dimensional state spaces using human feedback."
>
>
> ### 2. Questions regarding our experiment settings (why only two environments were used in some cases)
>
> We would start by explaining the intuition behind our experiment design:
> - Step 1: Use all 5 environments to fully evaluate EXPAND (with a synthetic oracle)
> - Step 2: Given limited computational resources and the fact that we already knew human explanatory feedback can improve DQN-Feedback across the five tasks, we pick a subset of the environments to perform additional experiments to get more insights into how saliency feedback benefits DQN-Feedback.
> - Step 3: Since EXPAND shows promising performance in Step 1 (experiments with oracle), we would like to see if the same conclusion can hold with human inputs. Considering that doing human experiments is much more expensive, we pick only one environment to train EXPAND with human teachers (we pick Enduro because its car racing scenario is closer to some real-world application).
>
> We will add this clarification to our revised version. To further address Reviewer 3’s concern, we conducted additional experiments on two other environments Enduro-1000 and Asterix. Due to limited time, for the EXPAND versus Context-Agnostic Data Augmentations experiment, we were only able to do it on the Enduro-1000 environment. The results can be found in the following anonymous link:
>
> https://anonymous.4open.science/r/EXPAND-neurips-2021-F1AC
>
> The additional results are consistent with the results from other environments in the paper:
> - In the ablation study, the advantage loss and invariance loss together yield the best results.
> - In EXPAND vs explanatory interactive learning methods, EXPAND outperforms Ex-AGIL and Attention-Align significantly. And again, Attention-Align fails to learn a usable policy.
>
> We will add these results to the paper/appendix.
>
>
> ### 3. Comparison to explanatory interactive learning methods (implementation errors?)
>
> We want to assure the reviewers of the correctness of our baseline implementations. In fact, we had several discussions with an author of AGIL [1] and Attention-Align [2], and we have largely reused the source codes from them. Both EXPAND and our baseline explanatory interactive learning methods are trying to solve a similar problem -- how to add human attention/saliency prior as an inductive bias to a complex deep learning system. Previous works only demonstrate the benefit of attention prior in supervised learning systems (e.g. imitation learning), and how to utilize the attention information remains an open question in DRL research (in fact this direction hasn’t gained much attention yet). As reported in Attention-Align, the weight of the attention regularization term might greatly affect the performance of an easier-to-stabilize supervised learning model.  We do agree that the performance of Attention-Align might get improved with a more careful hyper-parameter grid-search, but we would note that the inefficacy of Attention-Align is essentially highlighting a unique advantage of using data augmentation to regularize a brittle DRL model (over using the built-in attention to regularize the network).
>
> Regarding Ex-AGIL vs EXPAND, yes, one reason for EXPAND outperforming Ex-AGIL is EXPAND avoids the need to separately learn an attention mask model. Reviewer 3 asks if we are able to do a lot more augmentation when the learned mask model in Ex-AGIL is good. Here we argue that learning a good attention model can be even harder and slower than learning a good policy model, especially considering that under an interactive RL setting, human saliency samples can be much more sparse than environment samples (on the contrary, the original AGIL works on a supervised imitation learning setting, in which a large human attention dataset has to be collected beforehand).
>
> Regarding the "all three tasks" in line 374, we apologize that this is a typo and we will fix it in our revised version.
>
> [1] Zhang, Ruohan, et al. "Agil: Learning attention from human for visuomotor tasks."
>
> [2] Saran, Akanksha, et al. "Efficiently Guiding Imitation Learning Agents with Human Gaze."
>
> ### 4. Number/Percentage of frames got augmented
>
> In our experiment setting, all the algorithms (EXPAND and baselines) query the human/oracle every 4 episodes. Since the synthetic oracle evaluates and annotates every queried state, the percentage of augmented states is around 25% (e.g. all the algorithms ran 200k environment steps in Pong, so around 50k states got augmented). In human experiments, the average number of saliency feedback is 2026 (this number can be found in the paper), so the corresponding percentage is 4%. We will report this number in a clearer way in our revised version.
>
>
> ### 5. EXPAND versus Context-Agnostic Data Augmentations (why not use the augmentations on all states, if there is an oracle? 100% gaussian blur vs. gaussian blur except when there's human data)
>
> In reality, a human teacher can easily get exhausted, so the human feedback signals collected are typically much less than the environment samples. Since the synthetic oracle is used to simulate human teacher’s behavior, we only augment a small portion of all the states. Please note that, given the nature of human feedback, it’s very important to examine whether an interactive reinforcement learning method can work with limited human inputs.
>
> In the EXPAND vs Context-Agnostic Data Augmentation experiments, the specific hypothesis we wanted to test was whether our saliency-conditioned data augmentation is more informative than other standard data augmentations. Hence, in our experiments, we restricted both standard augmentation and context-aware augmentation to queried states.
>
> We agree that there can be many other interesting experiments to conduct (e.g. use context-aware augmentation and context-agnostic augmentation at the same time, and see what’s the best sample efficiency we can achieve), but we also believe that our existing experiments are revealing the most important features/insights of EXPAND. Hence, we gave higher priority to our current experiments, given there is a page limit and time limit. For the future version of this paper, we will try our best to conduct more complementary experiments.

---

> > ### Author Response · Authors · 2021-08-10
> > **Response to Review (continued)**
> >
> > ### 6. “Cutout” can be an alternative to “Gaussian Blur” in EXPAND
> >
> > Our choice of state augmentation method is based on the following considerations:
> > - The augmented states should look as realistic as possible
> > - The augmentation should not introduce new entities that have specific meaning in a domain. For example, in our Taxi domain, black block represents the destination of the taxi agent, so using “cutout” in Taxi might greatly alter the state context.
> > - The augmentation method should provide enough style variability to highlight relevant features and the irrelevant features. As an example, the Random-Crop transformation preserves most state information but provides very limited variability.
> >
> > Without compromising any factor too much, we find “Gaussian Blur” is the best choice. In fact, the use of Gaussian Blur is consistent with the conclusion in another Explainable RL paper [1] (though their paper uses Gaussian Blur in an inverse way: they use it to identify important features that the RL agent is “looking” at, while in our case, we use it to encourage the RL agent to “look” at the right regions).
> >
> > [1] Greydanus, Samuel, et al. "Visualizing and understanding atari agents."
> >
> >
> > ### 7. The human trainer seems to do worse than the synthetic oracle. Is this just because the oracle is quite good? Why DQN-TAMER is not in human experiments?
> >
> > The results from human experiments do not indicate the oracle is good or human feedback is worse. Human feedback-providing behavior can be very complex and cannot be fully simulated by an oracle. For instance, human teachers might give fewer feedback than an oracle (we allow the human trainer to skip a state), or they might give incorrect feedback (noise), but they might also provide better saliency feedback at states where salient regions were inaccurately hard-coded, etc. All these factors account for the result differences between oracle experiments and human experiments. But please note that EXPAND consistently outperforms DQN-Feedback baseline in both experiment configurations, suggesting that saliency feedback can robustly and effectively improve DRL sample efficiency regardless of how the feedback is collected.
> >
> > The main purpose of this paper is to show saliency feedback can effectively augment binary evaluative feedback. Hence, to control the variables in our human experiments, we used EXPAND and DQN-Feedback, which share the same underlying algorithm to learn from binary evaluative feedback. Please note that in all other experiments, DQN-Feedback even outperformed DQN-TAMER, indicating it’s a stronger baseline.

---

### Official Review · Reviewer_wPs5 · 2021-07-16

**Rating:** 7
**Confidence:** 4

**Summary:**

The paper proposes a way of data augmentation based on human feedback. Namely a human experts is tasked with marking on relevant areas of the state space (visual input, frames) with bounding boxes. The rest of the input is then blurred, which, presumably, should lead to the learning algorithms "focusing" on relevant areas and learning to ignore irrelevant (blurred) ones. Experiments on 5 simulated environment show that applying this technique leads to somewhat lower sample complexity and better final performance.

In my estimation this is an interesting idea on using human-in-the-loop for RL training and would be useful for RL practitioners working on such visual environment where data samples are expensive. Real-world robotics would be one such example.

**Limitations And Societal Impact:**

The limitations of the method are adequately addressed in the text of the manuscript.

**Main Review:**

Overall I do not have any specific questions or criticism or this work. The idea is valuable, the description is clear and the execution is valid and allows me to trust the results reported in the paper.

Originality: The idea to use humans to annotate the important regions of the input sample is quite original and in my estimation is well executed in this work, allowing the community to evaluate the benefits of the proposed trick. In general, human in the loop framework offers, in my opinion, a multitude of ways of enhancing the RL process and this work explores an interesting aspect of this area of study.

Quality: The claims are well-formulated and are supported by the empirical results provided.

Clarity: The proposed method is presented clearly and with a good level of technical detail allowing to understand the method down to the low-level technical details and mathematical formulation. Experiments and ablations are well-justified and address the right questions.

Significance: RL is known to be notoriously data hungry which is the main obstacle for its mass adoption in real physical systems where data samples are expensive. The proposed method add a potentially useful tool to the toolbox of RL practitioner working on application where sample complexity is crucially important. I find the contribution of this work to be significant enough to deserve being seen by the RL community. Also, while the result is presented in the context of human-in-the-loop, the Oracle experiments (Section 5.1) demonstrate the benefit of the proposed method also in a fully automated training process. As long as the environment allows to create a useful oracle, this method can be used to speed up learning as an additional module to any vision-based RL process.


**Time Spent Reviewing:**

4

---

> ### Author Response · Authors · 2021-08-10
> **Response to Review**
>
> We would like to thank the reviewer for their comments on our work. We feel more confident regarding the work with their positive feedback. Finally, we agree with their observation that EXPAND can indeed be helpful in cases when an automated oracle is available and the fact that the idea of using human/oracle explanations has high utility not only in speeding up the training procedure of an RL agent, but also potentially in the situation of performing necessary value alignment between the human and the agent.

---

### Official Review · Reviewer_o7pH · 2021-07-19

**Rating:** 7
**Confidence:** 3

**Summary:**

This paper proposes EXPAND - a method for allowing human input and semantic understanding to guide policy learning for RL agents. Humans are asked to provide binary feedback (good/bad) on actions taken by the agent, as well as highlight visual features which helped guide their decision (e.g. stop sign tells human to stop). The binary feedback is used to train an advantage function by leveraging a large margin classification loss. The visual highlighting is used to supervise feature learning by way of a consistency loss on perturbed data (i.e. blur segments not highlighted by human).

**Limitations And Societal Impact:**

I would ask the authors to more thoroughly discuss limitations of their method. In particular, if performance is very different between oracle and human input then that is important to discuss. Furthermore, the human input seems far more time consuming to produce. For example, the Enduro-1000 human experiment is only run to 60K steps v.s. 100K+ for oracle. The authors claim this takes total of 30 minutes, but if that's the case why not run for as many steps as the oracle?

Regarding social impacts, the authors claim EXPAND would help humans align the agent's behavior with our own value system. However, this paper does not show experiments where the RL reward is necessarily at odd with human values. Thus, this claim does not seem substantiated, and I would recommend the authors re-write the social impacts section accordingly.

**Main Review:**

It seems the primary difference between EXPAND and previous work DQN-TAMER [1] is that human input is integrated into an Advantage Function (which is also tuned w/ RL), rather than linearly blended with a separate Q function. This change does seem novel and results in a more elegant method with less hyper-parameters. The experimental results show EXPAND significantly outperforms DQN-TANNER in all environments. However, I do wish the authors more precisely described DQN-TAMER, and made these points clearer in the paper.

It's important to note that most experiments in this paper used an oracle to simulate human input likely to make the study less time consuming. The authors do one experiment using human input on the Enduro-1000 environment, but its difficult to tell if the performance matches the oracle Enduro-1000 runs due to separate y-axis scales. Furthermore, DQN-TAMER is only run with oracle input and not human. I hope the authors fix these bugs, so it becomes more clear if EXPAND can compete with DQN-TAMER when using human input.

Finally, the ablations do a good job justifying visual feature annotation helps with downstream RL task performance. I'm not convinced by the author's claim that Gaussian blurring somehow uniquely doesn't change state content. After all, what if high frequency information is important in the state? However, I do agree that it seems to work fine for the chosen environments.

[1] Arakawa, Riku, et al. "Dqn-tamer: Human-in-the-loop reinforcement learning with intractable feedback." arXiv preprint arXiv:1810.11748 (2018).

**Time Spent Reviewing:**

1.5

---

> ### Author Response · Authors · 2021-08-10
> **Response to Review**
>
> We thank the reviewer for their comments and suggestions, here are some clarifications on some of the questions raised:
>
> ### 1. Primary difference between EXPAND and previous work DQN-TAMER
>
> We agree that integrating human feedback into an Advantage function distinguishes EXPAND from DQN-TAMER. Apart from this, another significant difference is that EXPAND allows human teachers to give binary evaluative feedback and point out important visual features. And then, the context-aware state augmentation is used to effectively regularize the DRL model, thereby boosting the performance. We believe our main contribution is to demonstrate that richer/multi-modal human inputs can benefit conventional learning-from-binary-feedback interactive RL.
>
> We will revise the paper with a more detailed description of DQN-TAMER and its differences to EXPAND.
>
>
> ### 2. Oracle experiments vs human experiments, and the configuration of human experiments (50K steps v.s. 100K+ steps in oracle experiments)
>
> There are many possible factors that could account for the differences between oracle experiments and human experiments. One possible cause is human teachers typically give fewer feedback than an oracle (we allow the human trainer to skip a state). Also, humans might give incorrect/inconsistent feedback (noise), while they can provide better saliency feedback at states where salient regions were inaccurately hard-coded. Nevertheless, please note that EXPAND consistently outperforms the DQN-Feedback baseline in both experiment configurations, suggesting that saliency feedback can robustly and effectively improve DRL sample efficiency regardless of how the feedback is collected.
>
> Regarding why DQN-TAMER was not used in human experiments, please note that the central problem to answer in this paper is whether saliency feedback can effectively augment binary evaluative feedback. Hence, considering human experiments are more expensive and to control the variables in our human experiments, we used EXPAND and DQN-Feedback, which share the same underlying algorithm to learn from binary evaluative feedback. We would also like to refer the reviewers to the fact that DQN-Feedback (slightly) outperformed DQN-TAMER in all other experiments. So we were essentially comparing EXPAND to a stronger baseline.
>
> Regarding why the human experiments ran for 60k steps while the oracle experiments ran for 100k+ steps, this is because we were trying to answer different questions in these two experiments:
>
> - In oracle experiments, since the simulated feedbacks are less expensive, we didn’t restrict the access to the oracle and we wanted to see how fast can EXPAND and other baselines learn a good enough policy. The results suggest that to learn a similarly good policy, EXPAND takes significantly fewer environment samples and human feedback samples.
>
> - In human experiments, we instead wanted to know: i) within the same wall-clock time, can EXPAND still outperform the learning-from-binary-feedback baseline DQN-Feedback? Please note that for a human teacher, providing saliency feedback can take longer than providing binary judgment. To fairly compare EXPAND and DQN-Feedback, we take actual time cost into consideration and restrict the interaction time to 30 min. ii) Can EXPAND still work well given the complexity of human feedback (e.g. sparsity and noise)? iii) Can humans provide saliency feedback in an “effortless” way with the help of the object tracker? Please note that since the experiments showed the cost of providing visual explanation is low, we happened to see both EXPAND and DQN-Feedback were run for around 50k steps.
>
>
> ### 3. Gaussian blurring might alter state content
>
> We would like to clarify that we didn’t claim that Gaussian blurring doesn’t alter state content. Our statement was Gaussian blurring doesn’t introduce new “objects” into the images. We would like to explain some important factors that we considered when we chose the image transformation:
>
> - The augmented states should look as realistic as possible
> - The augmentation should not introduce new entities that have specific meaning in a domain. For example, in our Taxi domain, black block represents the destination of the taxi agent, so using “cutout” in Taxi might greatly alter the state context.
> - The augmentation method should provide enough style variability to highlight relevant features and irrelevant features. As an example, the Random-Crop transformation preserves most state information but provides very limited variability.
>
> Without compromising any factor too much, we find “Gaussian Blur” is the best choice. Actually, the use of Gaussian Blur is consistent with the conclusion in another Explainable RL paper [1] (though their paper uses Gaussian Blur in an inverse way: they use it to identify important features that the RL agent is “looking” at, while in our case, we use it to encourage the RL agent to “look” at the right regions).
>
> We will make sure to add clarification regarding this in our revised paper.
>
> [1] Greydanus, Samuel, et al. "Visualizing and understanding atari agents."
>
>
> ### 4. Limitations of EXPAND
>
> The limitations of EXPAND were discussed in the Future Work section. For instance, as pointed out by Reviewer 1, Gaussian Blur might still generate unrealistic states. Hence, one possible improvement would be replacing Gaussian Blur with some advanced state/image manipulation methods like [1]. Also, drawing bounding boxes is not the most natural way for the human to provide feedback. One improvement can be obtaining visual explanation and advice from natural language descriptions.
>
> [1] Park, Taesung, et al. "Semantic image synthesis with spatially-adaptive normalization."
>
>
> ### 5. Other comments
>
> Again, we thank the reviewer for all the comments, and we will make necessary changes to the paper (e.g. the social impact section) according to them.

---

### Author Response · Authors · 2021-08-10
**Response to Review (Additional Experiment Results)**

As requested by Reviewer 3 (Reviewer 1gtV), we conducted additional experiments (Ablation study, and EXPAND vs explanatory interactive learning methods) on two other environments Enduro-1000 and Asterix. The results can be found in this anonymous link:

https://anonymous.4open.science/r/EXPAND-neurips-2021-F1AC

The additional results are consistent with the results from other environments in the paper:
- In the ablation study, the advantage loss and invariance loss together yield the best results.
- In EXPAND vs explanatory interactive learning methods, EXPAND outperforms Ex-AGIL and Attention-Align significantly. And again, Attention-Align fails to learn a usable policy.

---

### Decision · Program_Chairs · 2021-09-27

**Decision:**

Accept (Spotlight)

**Comment:**

The paper presents a method for incorporating richer feedback than a reward signal from humans for training RL agents. Humans provide feedback by (i) indicating which actions are good/bad and (ii) highlighting salient areas in visual observations. All reviewers unanimously agree that the paper should be accepted. The experiments provided by the authors during the rebuttal phase further strengthen the paper. At the same time, I recommend the authors to improve the clarity of the paper, explanation of prior work (e.g., DQN-TAMER) and address other minor concerns raised by the reviewers in the camera-ready version.

If I were to simply go by the technical contributions of the paper, I would recommend it as a Poster. However, the line of research on providing human feedback beyond reward functions is worth highlighting to the research community. I therefore recommend the paper to be presented as a spotlight presentation.